# Coordination of robust single cell rhythms in the *Arabidopsis* circadian clock via spatial waves of gene expression

Peter D Gould[1†], Mirela Domijan[2,3†], Mark Greenwood[1,2,4], Isao T Tokuda[5], Hannah Rees[1], Laszlo Kozma-Bognar[6,7], Anthony JW Hall[1,8]*, James CW Locke[2,4,9]*

[1]Institute of Integrative Biology, University of Liverpool, Liverpool, United Kingdom; [2]Sainsbury Laboratory, University of Cambridge, Cambridge, United Kingdom; [3]Department of Mathematical Sciences, University of Liverpool, Liverpool, United Kingdom; [4]Department of Biochemistry, University of Cambridge, Cambridge, United Kingdom; [5]Department of Mechanical Engineering, Ritsumeikan University, Kusatsu, Japan; [6]Biological Research Centre, Hungarian Academy of Sciences, Szeged, Hungary; [7]Department of Genetics, University of Szeged, Szeged, Hungary; [8]Earlham Institute, Norwich Research Park, Norwich, United Kingdom; [9]Microsoft Research, Cambridge, United Kingdom

**\*For correspondence:**
anthony.hall@earlham.ac.uk
(AJWH);
james.locke@slcu.cam.ac.uk
(JCWL)

†These authors also contributed
equally to this work

**Competing interests:** The
authors declare that no
competing interests exist.

**Reviewing editor:** Richard
Amasino, University of
Wisconsin, United States

**Abstract** The *Arabidopsis* circadian clock orchestrates gene regulation across the day/night cycle. Although a multiple feedback loop circuit has been shown to generate the 24-hr rhythm, it remains unclear how robust the clock is in individual cells, or how clock timing is coordinated across the plant. Here we examine clock activity at the single cell level across *Arabidopsis* seedlings over several days under constant environmental conditions. Our data reveal robust single cell oscillations, albeit desynchronised. In particular, we observe two waves of clock activity; one going down, and one up the root. We also find evidence of cell-to-cell coupling of the clock, especially in the root tip. A simple model shows that cell-to-cell coupling and our measured period differences between cells can generate the observed waves. Our results reveal the spatial structure of the plant clock and suggest that unlike the centralised mammalian clock, the *Arabidopsis* clock has multiple coordination points.
DOI: https://doi.org/10.7554/eLife.31700.001

## Introduction

The circadian clock controls gene expression throughout the day and night in most organisms, from single cell photosynthetic bacteria to mammals (*Bell-Pedersen et al., 2005*; *Dunlap and Loros, 2017*). In many cases, a core circuit that generates this rhythm has been elucidated and been shown to oscillate in single cells. In multi-cellular organisms, these single cell rhythms can be integrated to allow a coordinated response to the environment (*Bell-Pedersen et al., 2005*). Mammals achieve this by driving oscillations in peripheral tissues from a central pacemaker in the brain, the suprachiasmatic nucleus (SCN) (*Pando et al., 2002*; *Reppert and Weaver, 2002*).

The *Arabidopsis* circadian clock generates a 24 hr rhythm in multiple key processes, including stomata opening, photosynthesis, and hypocotyl elongation (*Hsu and Harmer, 2014*). A hierarchical structure for the plant clock has recently been proposed, similar to that for the mammalian clock, where the shoot apex clock drives the rhythms in the root (*Takahashi et al., 2015*). However, there are further tissue-dependent differences that must be explained. For example, experiments using a

**eLife digest** The cycle of day and night sets the pace for the existence of most life on Earth. In response, many organisms have evolved internal biological clocks which are synchronized with the rhythm of light and dark. Thanks to this 'circadian clock', plants and animals can predict the onset of dawn and dusk, and schedule biological processes to specific times of the day.

Each cell possesses its own circadian clock, which is formed of a complex network of genes and proteins that get activated along a 24-hour cycle. This raises the question: how do these cells coordinate to create a body clock at the level of the entire organism? In mammals, this synchronisation happens thanks to a structure in the brain, but much less is known about how this occurs in plants. This is partly because few studies exist that focus on measuring the clocks of individual plant cells.

Here, Gould, Domijan et al. monitor the circadian clock of single cells in seedlings of a plant called *Arabidopsis thaliana* across several days and under constant environmental conditions. To do so, they use time-lapse microscopy and genetic methods to see when and where one of the clock's core genes is switched on.

The results show that, at the level of the plant, *Arabidopsis* has two waves of clock gene expression, one that goes up and one that goes down the root. In addition, the various parts of the plant have slightly different circadian rhythms – for instance, the tip of the root has a faster clock. Robust clock rhythms are also detected in individual cells across the plant. Clocks in neighbouring cells are found to communicate with each other to keep track of time, which might be contributing to this robustness. Mathematical simulations show that, when the individual clocks interact, they generate patterns of clock activity across the plant, which explains the two waves of gene expression in the root.

Plant circadian rhythms control traits that are crucial for agriculture, such as growth, yield, disease resistance and flowering time. Understanding, and ultimately controlling, the intricate cogs of these clocks may one day allow scientists to create better performing crops.

DOI: https://doi.org/10.7554/eLife.31700.002

luciferase reporter for clock activity have shown waves of clock gene expression in leaves (*Fukuda et al., 2007*; *Wenden et al., 2012*), as well as striped expression patterns in roots (*Fukuda et al., 2012*).

Beyond the coordination of plant rhythms, how robust the circadian clock is in individual cells across the plant is also unclear. Through integration of data from whole plant studies, a genetic circuit consisting of multiple coupled feedback loops has been proposed to generate the 24-hr rhythm (*Fogelmark and Troein, 2014*; *Pokhilko et al., 2012*). Simulations of this network display stable oscillations (*Figure 1a*), although experimental measurements of clock rhythms under constant conditions often display damped rhythms (*Figure 1b*) (*Gould et al., 2013*; *Locke et al., 2005*; *Locke et al., 2006*; *Salomé and McClung, 2005*). The damping is characterised by the reduction in amplitude of the oscillations through time. This damping at the whole plant level could be due to the clock circuit in all individual cells damping (top, *Figure 1c*), or to cells desynchronising due to different intrinsic periods or phases (*Guerriero et al., 2012*; *Komin et al., 2011*) (bottom, *Figure 1c*), or cells desynchronising due to stochasticity in clock activity (*Guerriero et al., 2012*). Previous studies have attempted to measure the clock in *Arabidopsis* at single-cell resolution; however, these have focused on specific parts of the plant for short time series (*Takahashi et al., 2015*; *Yakir et al., 2011*).

Here, using a reporter for the core clock gene *CIRCADIAN CLOCK ASSOCIATED 1* (*CCA1*) (*Wang and Tobin, 1998*), we examine the dynamics of the *Arabidopsis* clock across the whole plant at the single cell level over several days. Our results reveal that damping of rhythms at the whole plant level is mainly due to desynchronisation of oscillating single cells with different periods, and not due to noise in gene expression due to low molecule numbers or damping of individual oscillations. We observe two waves of clock gene expression, one up and one down the root. These waves, combined with more variable clock periods at each position in the root, cause the most desynchronisation. From our single cell data, we are able to estimate the coupling strength between cells, and

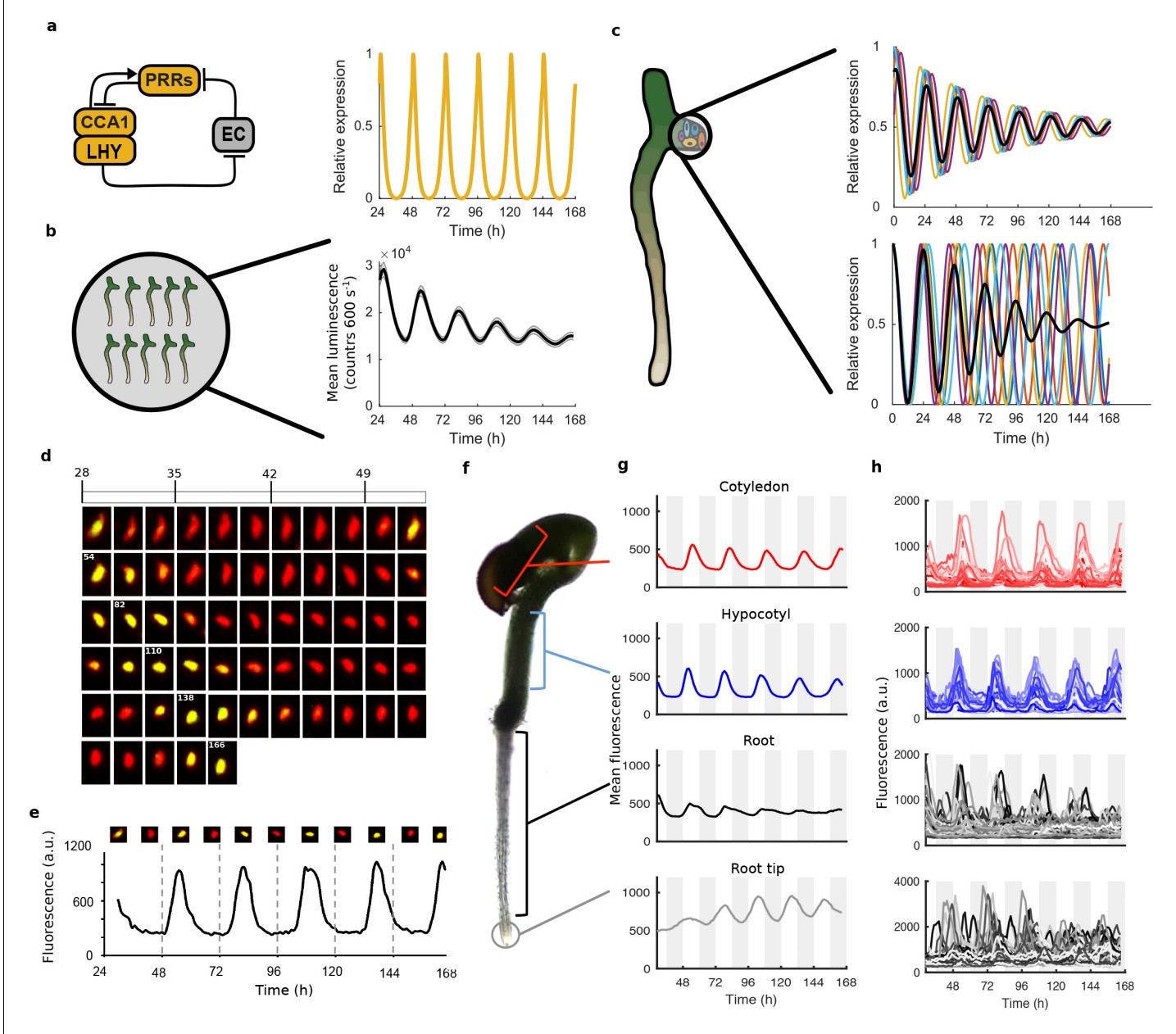

**Figure 1.** Quantitative time-lapse microscopy reveals single cell clock dynamics across the plant. (**a**), Current models of the clock (*Pokhilko et al., 2012*) predict undamped oscillations. (**b**), *CCA1:LUC* expression bulk averaged over multiple seedlings shows damping clock oscillations (mean ± s.e. m.; *n* = 32 seedlings). (**c**), The reason for damping could be due to damping rhythms in individual cells (top) or due to desynchronisation between cells (bottom). (**d**), Images show a representative nuclei from the cotyledon of a *cca1-11 CCA1*::CCA1-YFP; *35S*::H2B-RFP seedling that had been grown in LD cycles before being released into continuous light free running conditions for several days. The red channel represents when H2B fluoresce and the yellow CCA1. Times of peak expression are indicated on images. (**e**), Expression levels of CCA1-YFP from the representative nuclei shown in (**d**). Images of the nuclei are also shown for the peaks and troughs in the CCA1-YFP oscillation. (**f**), Representative seedling identifying the different sections imaged. (**g**), Mean traces of single cell CCA1-YFP for over 5 days of constant light in different regions of the plant showing damping rhythms in the root, but not the root tip. (**h**), CCA1-YFP traces from individual cell in each section for the same 5 days.
DOI: https://doi.org/10.7554/eLife.31700.003

The following source data and figure supplements are available for figure 1:

**Source data 1.** The percentage of rhythmic cells for WT experiment.
DOI: https://doi.org/10.7554/eLife.31700.010

**Source data 2.** The percentage of rhythmic cells for repeat WT experiment.
DOI: https://doi.org/10.7554/eLife.31700.011

*Figure 1 continued on next page*

*Figure 1 continued*

**Figure supplement 1.** The CCA1-YFP protein is functional and rescues the *cca1-11* mutation.

DOI: https://doi.org/10.7554/eLife.31700.004

**Figure supplement 2.** Confocal imaging and processing pipeline.

DOI: https://doi.org/10.7554/eLife.31700.005

**Figure supplement 3.** Mean and single cell oscillations of CCA1 nuclear localisation for repeat WT experiments.

DOI: https://doi.org/10.7554/eLife.31700.006

**Figure supplement 4.** Mean and single cell oscillations of CCA1 nuclear localisation for CCA1-long experiments.

DOI: https://doi.org/10.7554/eLife.31700.007

**Figure supplement 5.** Individual cell oscillations of CCA1-YFP reveal tissue specificity in robustness of oscillations.

DOI: https://doi.org/10.7554/eLife.31700.008

**Figure supplement 6.** Single cell rhythms have stable amplitudes across the plant.

DOI: https://doi.org/10.7554/eLife.31700.009

find evidence of coupling, especially strong in the root tip. A simple model suggests that our observed period differences, plus cell-to-cell coupling, can generate the observed waves in clock gene expression. Thus, our data reveals both the structure and robustness of the plant circadian clock system.

## Results

To analyse the dynamics of the plant clock at the single cell level, we constructed reporter lines that allowed us to quantitatively measure the nuclear level of the core clock protein CCA1 (*Wang and Tobin, 1998*). These reporter lines contained a CCA1-YFP protein fusion construct driven by the *CCA1* promoter in a *cca1-11* mutant background. They also contained a 35S::H2B-RFP nuclear marker to enable automatic detection of individual nuclei (*Federici et al., 2012*). By screening the clock phenotypes of multiple reporter lines we ensured that our reporter construct was functional and rescued the period phenotype of the *cca1-11* mutant (*Figure 1—figure supplement 1*). We took forward both a rescued wild-type period (WT) and a long period (CCA1-long) reporter line for further analysis.

We carried out time-lapse movies of *Arabidopsis* seedlings using a custom developed time-lapse confocal microscope setup (*Figure 1—figure supplement 2*). In order to examine the intrinsic behaviour of the clock we first entrained the seedlings to 12:12 hr light/dark cycles before examining the clock under constant conditions (constant blue light (30 µmol m$^{-2}$ s$^{-1}$) and temperature (22°C)). Our method allowed us to track and extract fluorescence values from the same individual nuclei over several days (*Figure 1d,e*). We first examined the average CCA1-YFP nuclear fluorescence signal from regions of the hypocotyl, cotyledon and roots (*Figure 1f,g*). We observed a robust oscillation in the cotyledon (red line, *Figure 1g*) and hypocotyl (blue line, *Figure 1g*), although with slight damping to the amplitude. In the top part of the root we observed strong damping of the circadian rhythm (black line, *Figure 1g*), although the oscillations recovered somewhat in the root tip (grey line, *Figure 1g*). Three repeat plants showed similar behaviour (*Figure 1—figure supplement 3*). We also observed similar behaviour in our CCA1-long reporter line (*Figure 1—figure supplement 4*), showing that our results remain true across a range of clock activity.

To determine the underlying cause of the damping in different tissues, we examined the clock rhythm in hundreds of cells from each seedling analysed (*Figure 1h*, *Figure 1—figure supplements 3b*, *4b* and *5*). Oscillations could be observed in all tissues, with most cells displaying circadian rhythms (see *Figure 1—source datas 1* and *2* for percentage of cells scored rhythmic for each plant section). Analysis of single cell oscillation amplitudes during the course of the movie revealed stable oscillations, with only a slight reduction in median oscillation amplitude in the root (*Figure 1—figure supplement 6*). Neither the strong damping at the tissue level in the root (*Figure 1g*), nor the damping observed at the plant level (*Figure 1a*), can be explained by individual cells losing rhythmicity (*Figure 1h*, *Figure 1—figure supplement 6*). We then examined the synchrony between cells and robustness of these rhythms in more detail. The hypocotyl and cotyledon were the most synchronised, with the amplitude of the mean trace nearly equalling the median amplitude of the individual cell time-series (*Figure 2a*, *Figure 2—figure supplements 1* and *2*). The hypocotyl and cotyledon rhythms also exhibited low period variability both within and between time traces, indicating a high

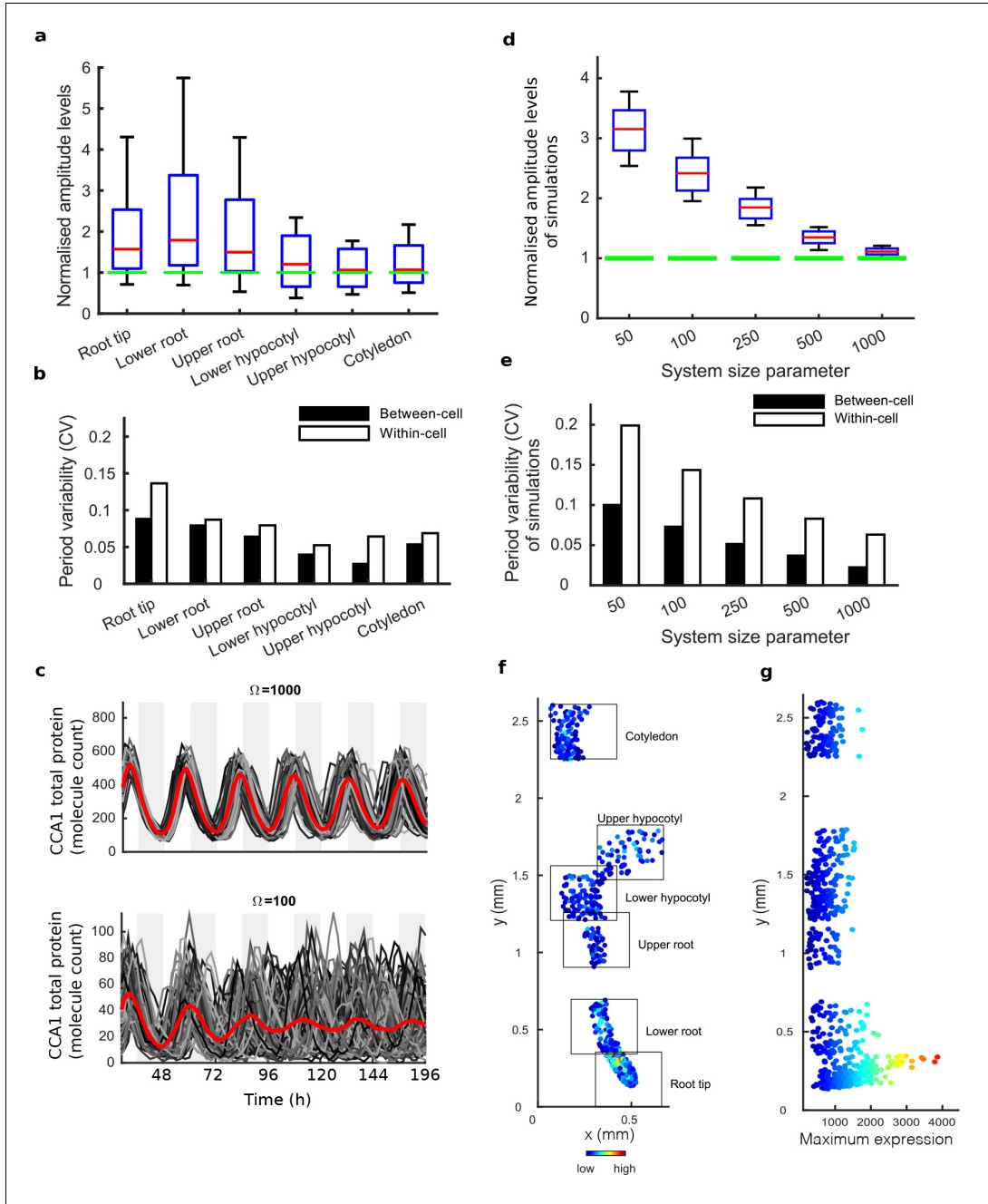

**Figure 2.** Single cell analysis reveals tissue level differences in robustness of the clock. (**a**), Rhythmic cell amplitudes in the imaged sections normalised to the amplitude of the mean trace (green line). Whiskers represent 9th and 91st percentile. (**b**), Between-cell and within-cell period variability in each imaged section. (**c**), Stochastic model CCA1 total molecule count for $\Omega = 1000$ (top) and $\Omega = 100$ (bottom) for 100 simulated runs (grey) plotted from 29 to 168 hr in constant light (comparable to the data in *Figure 1*). Means of all simulated runs are shown in red. $\Omega$ represents the system size. (**d**), Rhythmic simulated run amplitudes for different system sizes ($\Omega$) normalised to the mean simulation (green line). (**e**), Between and within cell variability of each simulation with different system size. (**f**), Scatterplot of the rhythmic cells in all imaged plant sections stitched together. Colour indicates the maximum expression. (**g**), Scatterplot of the maximum expression values vs. longitudinal position on the plant measured from the root tip. Colour legend is the same as (**f**). For root tip, $n = 242$; lower root, $n = 84$; upper root, $n = 46$; lower hypocotyl, $n = 114$; upper hypocotyl, $n = 53$; cotyledon, $n = 103$ ($n$ = number of rhythmic cells).
DOI: https://doi.org/10.7554/eLife.31700.012

The following figure supplements are available for figure 2:

*Figure 2 continued on next page*

*Figure 2 continued*

**Figure supplement 1.** Tissue level differences in robustness synchronisation, and period of single cell clock oscillations in repeat WT experiment.

DOI: https://doi.org/10.7554/eLife.31700.013

**Figure supplement 2.** Tissue level differences in robustness, synchronisation, and period of single cell clock oscillations in CCA1-long experiment.

DOI: https://doi.org/10.7554/eLife.31700.014

**Figure supplement 3.** Periods display no spatial structure in *z* direction for WT repeat and CCA1-long experiments.

DOI: https://doi.org/10.7554/eLife.31700.015

level of robustness (*Figure 2b*, *Figure 2—figure supplements 1b* and *2b*). However, the root displayed significant desynchronisation, with the amplitude of the mean trace lower than the median amplitude of the individual cell traces (*Figure 2a*, *Figure 2—figure supplements 1a* and *2a*), with higher variability in period within and between single cell lineages (*Figure 2b* and *Figure 2—figure supplements 1b* and *2b*).

We next asked whether the damping of amplitude in the mean rhythm is caused by noise in gene expression due to low clock molecule numbers, as proposed by previous stochastic modelling (*Guerriero et al., 2012*). Simulations of the stochastic model of the *Arabidopsis* gene network (*Guerriero et al., 2012*) display greater desynchronisation as the system size, effectively the simulated number of molecules in the cell, is reduced (*Figure 2c–e*). The lower the molecule number, the more desynchronisation at the single cell level, and the more damping of the mean expression (red line, *Figure 2c*), even though the individual traces do not display a damped rhythm. To explore this further, we simulated the model for a range of system sizes (*Figure 2c*) and examined the synchrony and robustness of the simulations (*Figure 2d,e*). Previously the system size was estimated to be on the order of 100 molecules per cell (bottom panel, *Figure 2c*) in order for the desynchronisation observed in whole plant measurements to be explained solely by noise (*Guerriero et al., 2012*). We compared the level of synchrony and period variability from simulations of this model to our experimentally measured single cell rhythms. The single cell rhythms that we detect in the hypocotyl and cotyledon have a lower period variability (*Figure 2b*) than a system size of 100 and in fact behave closer to a system size of 1000 (*Figure 2e*). The hypocotyl and cotyledon also have a smaller difference between the amplitude of the mean trace (green line) compared to the median amplitude of the individual cell time-series (red line) (*Figure 2a*) than a system size of 100 (*Figure 2d*), and again behaves closer to a system size of 1000 (*Figure 2d*). The level of desynchronisation observed in the root and root tip, however, behave more like a simulated system size of 100 (*Figure 2a–e*). Similar CCA1-YFP expression levels are observed across the plant (*Figure 2f,g*), so it is unlikely that this lack of robustness is due solely to a change of total molecule number for the clock system in different parts of the plant. In fact the amplitude and expression levels of the single cell oscillations in the root tip are high compared to other sections across the plant (*Figure 2f,g* and *Figure 2—figure supplements 1a,c–d* and *2a,c–d*).

If the strong damping in the root is not a result of noise due to low clock molecule numbers, or individual cells losing rhythmicity, what is the cause? The measured variable and desynchronous rhythms in the root suggest that the clock could be behaving differently in different parts of the root. To test this possibility, we plotted the period of the individual cell oscillations across the plant (*Figure 3a,b* and *Figure 2—figure supplements 1e,f and* and *2e,f*). We observed surprising spatial structure to the clock in the root. The upper sections of the root displayed longer periods than the rest of the plant, as reported previously for the whole root (*James et al., 2008*). However, we observed very fast rhythms in the root tip (*Figure 3a,b*), which is also the section with very high expression rhythms (*Figure 2f,g*). Although we do not observe evidence of phase resetting in the root tip, as proposed in an earlier luciferase study (*Fukuda et al., 2012*), this could be due to our different growth conditions and stage of plant development. A repeat plant showed similar results (*Figure 2—figure supplement 1c–f*), as did the CCA1-long reporter line (*Figure 2—figure supplement 2c–f*). Each radial section of the plant along the longitudinal axis can be made up of multiple cell types. Therefore, we next tested whether the rhythms have any spatial structure in the *z* direction, which would suggest that different cell types have different period rhythms. Plots of period in the *z*

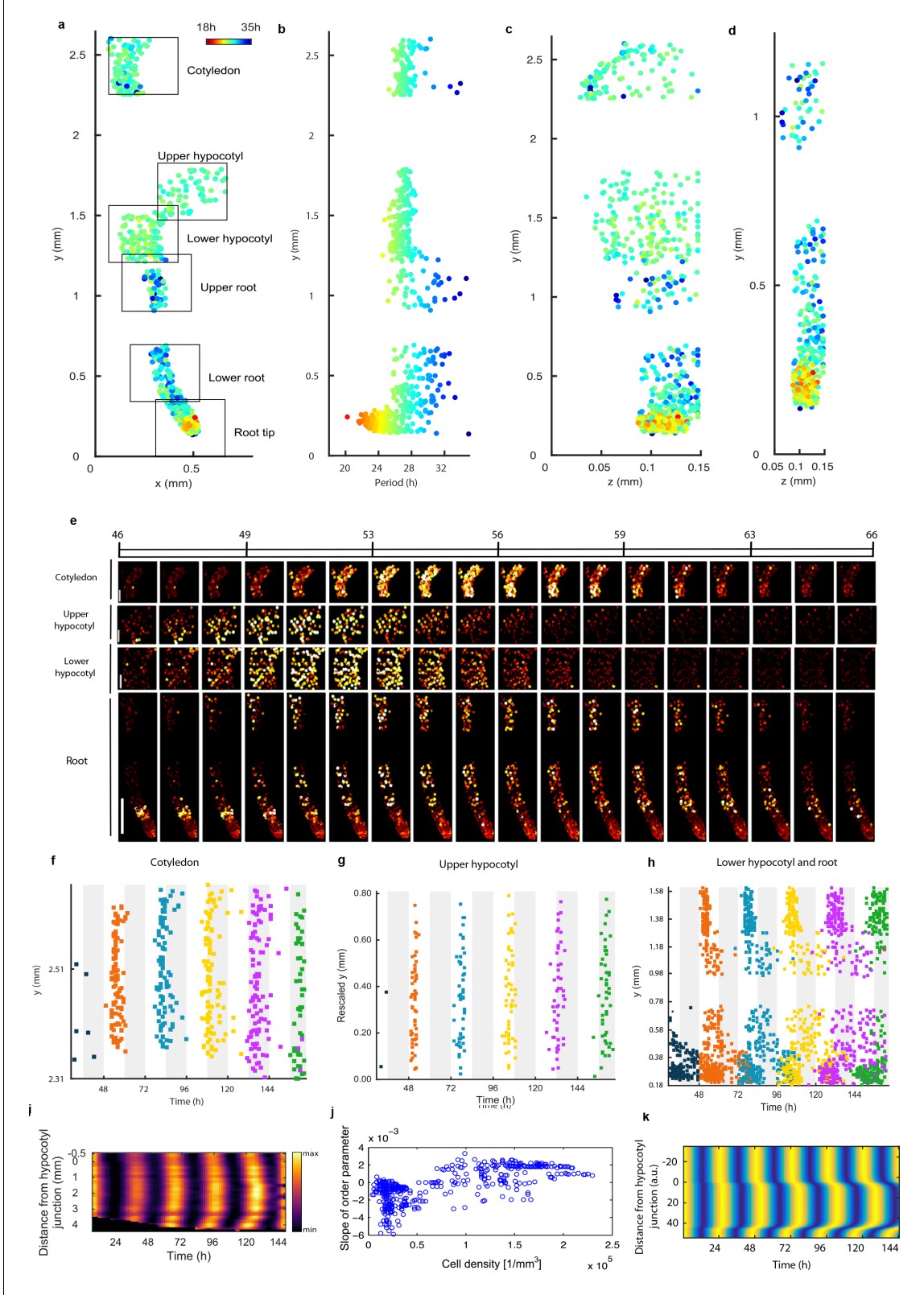

**Figure 3.** Single cell period differences and cell-to-cell coupling generate spatial waves of clock gene expression. (a), Scatterplot of the rhythmic cells in all imaged plant sections stitched together in *x-y* direction. Colour indicates the oscillation period. (b), Period values vs. longitudinal position on the plant measured from the root tip. Colour legend is the same as (a). (c), (d), Scatterplot in the *y-z* direction of rhythmic cells in all imaged plant sections (c) or in the root and root tip sections only (d). Colour legend is the same as (a). (e), Montage of the normalised CCA1-YFP expression of rhythmic cells

*Figure 3 continued on next page*

*Figure 3 continued*

from the root (bottom panel, first image taken after 46.1 hr in LL), lower hypocotyl (taken after 46.6 hr in LL), upper hypocotyl (taken after 46.7 hr) and cotyledon (top panel, taken after 46.9 hr in LL). Each frame is approximately 1.1 hr apart. Scale bar represents 0.25 mm for root panel and 0.1 mm for other panels. Both colour intensity and spot size indicate expression level. By colour, red indicates low and yellow high intensity. (f–h), Space-time plots of peak times of rhythmic cells across sections: cotyledon (f) upper hypocotyl (g), lower hypocotyl and root (all sections) (h). (i), Representative space-time plot of normalised *PRR9:LUC* expression across longitudinal sections of a seedling (N = 2, n = 7). *N* represents the number of independent experiments, *n* the total number of individual seedlings. Seedlings were imaged under constant red and blue light (Methods). (j), Slopes of the order parameter are plotted against cell densities. A positive slope indicates that the level of synchrony increases in time due to cell-cell interactions. (k), Space-time plot of simulated total normalised *CCA1* expression across longitudinal sections of the seedling.

DOI: https://doi.org/10.7554/eLife.31700.016

The following figure supplements are available for figure 3:

**Figure supplement 1.** Spatial waves of clock gene expression are seen in clock luciferase reporter lines.
DOI: https://doi.org/10.7554/eLife.31700.017
**Figure supplement 2.** Synchronisation analysis based on the order parameter.
DOI: https://doi.org/10.7554/eLife.31700.018
**Figure supplement 3.** Synchronisation analysis based on the synchronisation index.
DOI: https://doi.org/10.7554/eLife.31700.019
**Figure supplement 4.** Synchronisation analysis applied to CCA1-long dataset based on the order parameter.
DOI: https://doi.org/10.7554/eLife.31700.020

direction in each section do not reveal any discernible pattern (*Figure 3c,d*, *Figure 2—figure supplement 3*). This suggests that the differences in rhythms we observe are not restricted to a specific cell type.

To further understand the spatial structure of the clock, we examined montages of clock gene expression (*Figure 3e*; *Figure 2—figure supplement 1g*; *Videos 1–3*) and timing of peaks of expression (*Figure 3f–h*) across the plant. Examination of the clock phase between 48 and 72 hr reveals that the clock peaks earlier in the hypocotyl (mean phase in upper hypocotyl 51.2 ± 1.1h, mean phase in lower hypocotyl 52.1 ± 1.5h) than in the cotyledon (mean phase 55.6 ± 2.3h) (*Figure 3f–h*). From the top of the root the phase of the clock is shifted to later in the day as you go down the root (upper root mean phase 56.6 ± 4.5h) (*Figure 3h*). However, from the root tip (mean phase 56.2 ± 5.0h) the phase of the clock is shifted to later in the day as you go up (lower root phase 58.0 ± 4.4h). This generates two waves in the montages of clock gene expression, one going up and

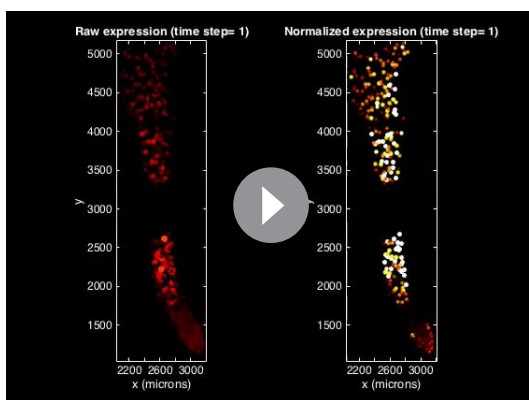

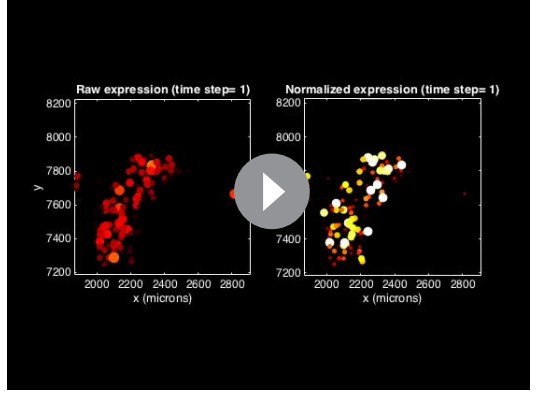

**Video 1.** Peaks of CCA1-YFP expression in the lower hypocotyl and root region. Video of CCA1-YFP raw (left panel) and normalised (right panel) expression in rhythmic cells imaged from 29 h (root section) or 29.5 h (lower hypocotyl) in LL. Both colour intensity and size of spot indicate expression level. Frame number is indicated by the time step and each frame is approximately 1.1 h apart.

DOI: https://doi.org/10.7554/eLife.31700.021

**Video 2.** Peaks of CCA1-YFP expression in the cotyledon region. Video of CCA1-YFP raw (left panel) and normalised (right panel) expression in rhythmic cells imaged from in LL. Both colour intensity and size of spot indicate expression level. Frame number is indicated by the time step and each frame is approximately 1.1 hr apart.

DOI: https://doi.org/10.7554/eLife.31700.022

one going down the root (*Figure 3e*; *Video 1*; *Figure 2—figure supplements 1g* and *2g*). At each *y* position in the root, the phases are also more variable than in the rest of the plant, with a standard deviation of phases of ±5.0 hr in the root tip compared to ±1.1 hr in the hypocotyl for phases occurring between 48 and 72 hr (*Figure 3h*). Thus, the damped mean rhythm of clock activity we observed in the root (*Figure 1g*) is caused by the averaging of these two waves of gene expression that have a broad distribution of phases. We observed qualitatively similar waves in luciferase reporter lines for the clock genes *PSEUDO-RESPONSE REGULATOR 9* (*Ito et al., 2003*) (*Figure 3i*; *Video 4*), *CCA1* (*Figure 3—figure supplement 1a*) and *GIGANTEA* (*Park et al., 1999*) (*Figure 3—figure supplement 1b*), suggesting that our results are a general property of the clock.

The coherent waves of gene expression suggested that the plant clock signal could be coupled. To estimate this coupling, we calculated the order parameter (*Kuramoto, 1984*) from our single cell data and estimated the coupling strength based on a technique developed for mammalian circadian cells (*Rougemont and Naef, 2007*). We observed signs of coupling across the plant, with the strongest evidence for coupling in the root tip, where the order parameter actually increased with time (*Figure 3j* and *Figure 3—figure supplement 2–4*). Interestingly, this occurred where cell density was highest, as observed in cultured SCN cells (*Aton et al., 2005*). To investigate the mechanism for the waves of clock gene expression in the root we developed a simple mathematical model where the cells are described by coupled oscillators with different periods, as informed by the data (*Figure 3k*). The periods of the clock in the model were faster in the shoot and root tip than the rest of the root, as measured experimentally. This simple model can generate waves of gene expression up and down the root that produce a bow wave in the space-time plot (*Figure 3k*), similar to that observed experimentally (*Figure 3h,i*).

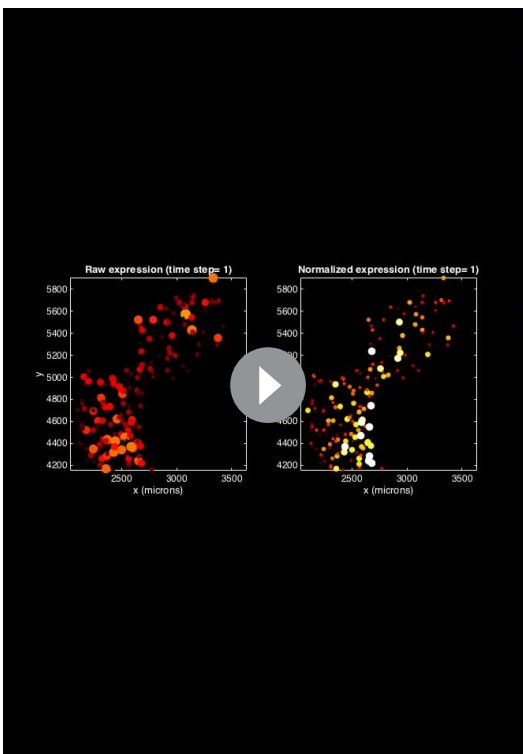

**Video 3.** Peaks of CCA1-YFP expression in the upper hypocotyl region. Video of CCA1-YFP raw (left panel) and normalised (right panel) expression in rhythmic cells imaged from 29 h in LL. Both colour intensity and size of spot indicate expression level. Frame number is indicated by the time step and each frame is approximately 1.1 h apart.
DOI: https://doi.org/10.7554/eLife.31700.023

## Discussion

Our single cell measurements have revealed tissue specific differences in the phases and robustness of the clock in *Arabidopsis*. These differences are not restricted to one cell type, as similar periods are observed in the *z* dimension through the plant (*Figure 3c,d*), suggesting that cells are instead responding to information based on their longitudinal position. The observed robust rhythms in the hypocotyl that peak before the cotyledon and the top of the root, with a wave of clock gene expression down the root, fit with a proposed hierarchical

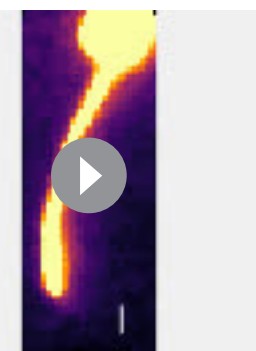

**Video 4.** Waves of *PRR9:LUC* expression in the lower hypocotyl and root. Video of *PRR9:LUC* luminescence in the hypocotyl and root of a single seedling from 8–144 h after transfer to constant light. Frame intervals are 1.5 h and scale bar shows 0.5 mm.
DOI: https://doi.org/10.7554/eLife.31700.024

structure for the plant clock, where the shoot apex clock drives the rhythms in the root (*Takahashi et al., 2015*). However, our results suggest that the structure of the plant clock is more complicated, as this hierarchical model does not explain the observed short period oscillations in the root tip, or the wave of clock gene expression up from the root tip. Our results support a more decentralised model of clock coordination in plants (*Endo, 2016*; *Endo et al., 2014*).

Earlier studies of the *Arabidopsis* clock argue either that cellular oscillations are uncoupled (*Thain et al., 2000*; *Yakir et al., 2011*), or weakly, but detectably, coupled (*Fukuda et al., 2007*; *Fukuda et al., 2012*; *James et al., 2008*; *Takahashi et al., 2015*; *Wenden et al., 2012*). Coupling has also been detected in *Lemna gibba* fronds, as examination of a clock bioluminescence reporter with single cell resolution showed weak coupling between individual clock oscillators under constant conditions (*Muranaka and Oyama, 2016*). These heterogeneous oscillations could be synchronised by light-dark cycles (*Okada et al., 2017*). Our single cell approach is consistent with weak coupling across the whole plant but reveals regions with strong local coupling between cells, especially in the root tip, which is sufficient to drive an increase in synchrony with time. In the future, it will be interesting to examine whether the shoot apical meristem, which has a high cell density like the root tip and has been shown to be coupled when excised from the plant, has a similarly high coupling strength given its role in driving rhythms in the shoot (*Takahashi et al., 2015*).

Our modelling reveals that cell-cell coupling together with the observed period differences between tissues is sufficient to replicate the decentralised spatial structure of the clock that we observe experimentally. Our simple model offers a qualitative match to data. In the future it will be interesting to develop more mechanistic models of the plant circadian clock at the single cell level, as has been done at the bulk level previously (*Fogelmark and Troein, 2014*; *Guerriero et al., 2012*; *Locke et al., 2005*; *Locke et al., 2006*; *Pokhilko et al., 2012*). One barrier to achieving this is that currently our estimations of clock periods from single cells are from the intact plant, and so will include the effects of cell-cell coupling. To develop mechanistic models at the single cell-level, it will be critical to develop reliable methods to measure the rhythmic properties of physiologically representative plant cells in physical isolation. It will also be important to test future models by examining the clock experimentally under a range of environmental conditions and at different developmental stages.

Decentralised coordination could create flexibility and allow parts of the plant to respond differentially to environmental perturbations. There is already evidence that the root clock may respond differently to light than the shoot (*Bordage et al., 2016*; *Nimmo, 2018*), and that the vasculature and epidermal clock regulate distinct physiological processes (*Shimizu et al., 2015*). It has been recently shown that initiation of lateral roots triggers the resetting of the clock in the emerging lateral root (*Voß et al., 2015*). In the case of lateral roots, auxin is proposed to be involved in resetting the clock (*Voß et al., 2015*). Waves of auxin also occur in the root (*Moreno-Risueno et al., 2010*), and it will be interesting to examine if the circadian clock wave is linked to these auxin dynamics. An important next step will be to investigate what the coupling signal is for the plant circadian clock (*Covington and Harmer, 2007*; *Dalchau et al., 2011*; *Haydon et al., 2013*).

## Materials and methods

### Key resources table

| Reagent type (species) or resource | Designation | Source or reference | Identifiers | Additional information |
|---|---|---|---|---|
| gene (*Arabidopsis thaliana*) | CCA1 | PMID: 9657153 | TAIR:AT2G46830 | |
| gene (*Arabidopsis thaliana*) | PRR9 | PMID: 14634162 | TAIR:AT2G46790 | |
| gene (*Arabidopsis thaliana*) | GI | PMID: 17102804 | TAIR:AT1G22770 | |
| strain, strain background (*Arabidopsis thaliana*) | cca1-11 | PMID: 14555691 | TAIR:1008081946 | |

*Continued on next page*

*Continued*

| Reagent type (species) or resource | Designation | Source or reference | Identifiers | Additional information |
|---|---|---|---|---|
| strain, strain background (*Agrobacterium tumefaciens*) | GV3101 | doi:10.1007/BF00331014 | | |
| transfected construct | *CCA1::CCA1-YFP* | This paper | Ask for 'CCY_pPB' | promoter::protein construct |
| transfected construct | *35S*::H2B-RFP | PMID: 22466793 | | promoter::protein construct |
| biological sample (*Arabidopsis thaliana*) | *CCA1::LUC* | PMID: 20530216 | Ask for 'B8-5' | promoter::luciferase construct; Col-0 background |
| biological sample (*Arabidopsis thaliana*) | *GI::LUC* | PMID: 20530216 | Ask for 'A2-1-4' | promoter::luciferase construct; Col-0 background |
| biological sample (*Arabidopsis thaliana*) | *PRR9::LUC* | PMID: 20530216 | Ask for 'G8-5' | promoter:Luciferase construct; Col-0 background |
| biological sample (*Arabidopsis thaliana*) | CCA1-YFP WT (Ws) | This paper | Ask for '1–1' | *CCA1*::CCA1-YFP; *35S*::H2B-RFP, WT clock period |
| biological sample (*Arabidopsis thaliana*) | CCA1-YFP long (Ws) | This paper | Ask for '3–1' | *CCA1*::CCA1-YFP; *35S*::H2B-RFP, long clock period |
| Recombinant DNA reagent | pPCV812 | PMID: 18980642 | | |
| sequence-based reagent | CCA1_CDS_Fwd | Sigma-Aldrich | | 5'-AAAGGATCCATGGAGACAAATTCGTCTGGA-3' |
| sequence-based reagent | CCA1_CDS_Rev | Sigma-Aldrich | | 5'-ATACCCGGGTGTGGAAGCTTGAGTTTCCAA-3' |
| sequence-based reagent | CCA1_prom_Fwd | Sigma-Aldrich | | 5'-AAAGAATTCATTTAGTCTTCTACCCTTCATGC-3' |
| sequence-based reagent | CCA1_prom_Rev | Sigma-Aldrich | | 5'-ATAGGATCCCACTAAGCTCCTCTACACAACTTC-3' |
| software | Imaris | BitPlane, Switzerland | | version 7.0 |
| software | ImageJ | National Institutes of Health, U.S.A. | | public domain |
| software | MATLAB | MathWorks, U.K. | | version 2015b |
| algorithm | MATLAB code | This paper | | https://gitlab.com/slcu/teamJL/Gould_etal_2018 |

## Constructs

The CCA1::CCA1-YFP was constructed as follows: The coding region of *CCA1* was amplified from *Arabidopsis* (Ws) genomic DNA using the primer pair CCA1_CDS_Fwd (5'-AAAGGATCCATGGAGA-CAAATTCGTCTGGA-3') and CCA1_CDS_Rev (5'-ATACCCGGGTGTGGAAGCTTGAGTTTCCAA-3'). The *CCA1* promoter region was amplified from *Arabidopsis* (Ws) genomic DNA using the primer pair CCA1_prom_Fwd (5'-AAAGAATTCATTTAGTCTTCTACCCTTCATGC-3') and the CCA1_prom_-Rev (5'-ATAGGATCCCACTAAGCTCCTCTACACAACTTC-3'). Unique restriction sites were designed at the ends of the amplicons to facilitate cloning. The fragment of *CCA1* coding region was cloned in the modified pPCV812 binary plasmid (*Pfeiffer et al., 2009*) between the *35S* promoter and the *YFP* gene via *Bam*HI (5') and *Sma*I (3') sites, resulting in *35S*::CCA1-YFP. Next, the *35S* promoter was replaced by the *CCA1* promoter fragment via *Eco*RI (5') and *Bam*HI (3') sites, resulting in CCA1::CCA1-YFP. The cloned *CCA1* promoter fragment was 862 bp in length and contained the full 5' - untranslated region, but not the ATG. The *35S*::H2B-RFP construct was described previously in (*Federici et al., 2012*). The *PRR9:LUC*, *CCA1:LUC* and *GI:LUC* reporters are in the Col-0 background and were also described previously (*Palágyi et al., 2010*).

## Plant growth material

The *CCA1::CCA1-YFP* construct was transformed in *cca1-11* (Ws) mutant background (*Hall, 2003*). Homozygous T3 generations of several independent transgenic lines were checked for

complementation via delayed fluorescence (*Gould et al., 2009*) (*Figure 1—figure supplement 1b, c*). The *CCA1*::CCA1-YFP expressing line showing full complementation was then re-transformed with *35S*::H2B-RFP which was used for tracking purposes during analysis (*Federici et al., 2012*).

*Arabidopsis* seeds were surface sterilised and suspended in 0.1% top agar and placed in 4°C for 2 days. After sowing, seeds were grown inside of growth incubators (Sanyo MLR-352) under 12:12 LD cycles in 80 umol m$^2$ s$^{-1}$ cool white light at 22°C for entrainment. Henceforth, these conditions are referred to as entrainment conditions.

## Luciferase and delayed fluorescence bulk imaging

10-20 seeds were sown onto Murashige and Skoog (MS) 2% agar in clear 96-well microtitre plates with at least 8 wells per line. A second clear microtitre plate was placed on top of the plate containing the seed to increase well height. These were then sealed using porous tape (Micropore). Seedlings were grown for 9 days under entrainment conditions and transferred to experimental conditions on dawn of the 10$^{th}$ day. For luciferase experiments, on the 9$^{th}$ day seedlings were sprayed with a 5 mM luciferin solution in 0.001% Triton x-100 before transfer to experimental conditions on dawn of the 10$^{th}$ day.

Imaging was carried out in Sanyo temperature controlled cabinets (MIR-553 or MIR-154) at 22°C and under an equal mix of red and blue LEDs (40 μmol m$^{-2}$ sec$^{-1}$ total). Seedlings were imaged using an ORCA-II-BT (Hamamatsu Photonics, Japan) or LUMO CCD camera (QImaging, Canada). Experiments were run over several days with images being taken every hour as described previously (*Gould et al., 2013*; *Litthauer et al., 2015*) Image analysis was carried out using Imaris (Bitplane, Switzerland) or ImageJ (NIH, USA).

## Luciferase macro imaging

Seedlings were sown in a row of eight seedlings on 2% agar media supplemented with MS media. Seedlings were grown upright for four days under entrainment conditions. Imaging commenced on dawn of the fifth day. Imaging was performed inside of Sanyo plant growth incubators at 22°C under an equal mix of red and blue light emitting diodes (40 μmol m$^{-2}$ sec$^{-1}$ total) (*Figure 3i*, *Figure 3—figure supplement 1a*) or under blue light emitting diodes (40 μmol m$^{-2}$ sec$^{-1}$ total) (*Figure 3—figure supplement 1b*). Seedlings were imaged upright using a LUMO CCD camera (QImaging, Canada). Experiments were run over several days with images being taken every 90 min.

## Confocal microscopy

For confocal experiments seed were sown directly onto glass bottom dishes (Greiner, Austria) in an array format. Once dry, the seeds were covered with 5 ml MS 2% agar media in absence of sucrose. Once set, dishes were sealed with porous tape (Micropore) and grown upright under entrainment conditions for 4 days. After 4 days plates were ready for imaging.

The microscopy pipeline is outlined in *Figure 1—figure supplement 2*. Up to 18 seedlings were grown in an array format on glass bottom dishes. At dawn on the fourth day of growth the dishes were fixed into the confocal temperature controlled stage (22°C) using the dish manifold. To maintain correct light conditions (30 μmol m$^{-2}$ s$^{-2}$ constant blue light) a custom-made light emitting diode (LED) rig was used. Growth conditions allowed slow growth during the movie, which enabled easier tracking of single cells. A Zeiss 710 (Zeiss, Germany) inverted confocal microscope with a 40x/ 1.2 water corrected oil objective was used for all imaging. YFP and RFP excitation was produced using a 514 nm laser and a main beamsplitter (MBS) 458/514. The 514nm laser was set to 4% power for all experiments. To reduce problems with auto-fluorescence and improve signal to noise ratio a lambda scan was carried using a ChS PMT and filters 492-658. Brightfield (BF) used a ChD PMT. Imaging was carried out using a 0.6 zoom to increase field of view. A motorised stage was used to allow multiple positions to be imaged across the plant per experimental run. The diameter of nuclei in our seedlings ranges in size from 6 μm (root tip) to 15 μm (hypocotyl). A resolution of 2 μm in the *z* dimension was chosen to allow the capture of several slices through each nucleus. Data were auto saved during imaging with data split into files by position imaged.

## Processing confocal images

Firstly blank images created by time-lapse being terminated early were removed in ImageJ. Also with ImageJ, lambda scans produced during confocal imaging were split into YFP (511 to 547 nm), RFP (586 to 625 nm) and brightfield (BF) spectrums and then reduced in dimensionality to give one channel for each wavelength. Data were then saved as OME TIFF, writing each time point as a separate file. Once processed, all the data were loaded into Imaris (Bitplane, Switzerland) and merged to produce one file containing YFP, RFP and BF. A median filter size 3x3x1 was then applied across all of the data. Detection of YFP/RFP expressing cells was carried out using the spot detection feature and tracking of spots over time was carried out using an autoregressive motion model using an estimated cell $x,y$ diameter of 6-10 µm. Data were then exported in excel format for further analysis. Details are provided in the subsequent section. Quality control checks were carried out at multiple points (see *Figure 1—figure supplement 2*). The first quality check was made to ensure that the seedling remained in the focal plane during the course of the experiment. If not, the dataset was not carried forward for further analysis. The second check was to make sure that no mistakes were made in the processing of the images (described above). This was carried out by examining the data in Imaris and ensuring that RFP signal was detected throughout the movie. The third check was carried out to correct any errors in tracking cells across the time-lapse data. The fourth check used the videos to more closely monitor the data for anything that looked problematic. The final check used the graphs to identify any problems that may have occurred during the whole single cell pipeline. If the laser power was not found to be stable during the course of the imaging, which could be observed by fluctuations in the background fluorescence, the dataset was not carried forward for further analysis.

## Single cell data processing

Period analysis was carried out in BioDare, an online system for data sharing and analysis (*Costa et al., 2013*; *Moore et al., 2014*). Since most of the period analysis methods in BioDare require evenly spaced time series, the data were first interpolated (using MATLAB's (MathWorks, U. K.) interp1 function and spacing of 1 hr). Period estimates were obtained by three different methods: Spectrum Resampling (*Costa et al., 2013*), FFT-NLLS (*Johnson and Frasier, 1985*; *Straume et al., 2002*) and mFourFit (*Edwards et al., 2010*). Cells were classed as rhythmic only if each method identified them as rhythmic (i.e. BioDare did not ask to ignore them), their goodness of fit was below one for FFT-NLLS and mFourfit or 0.9 for Spectrum Resampling and all estimates obtained by different methods were within 2.5 hr of each other. In *Figure 3*, *Figure 2—figure supplements 1* and *2*, the FFT-NLLS period estimates are shown. Period variability within and between cells was calculated as described previously (*Kellogg and Tay, 2015*).

Since some of the sections imaged overlap (e.g. *Figure 2f*; *Figure 3a*; *Figure 2—figure supplements 1c* and *2c*), in order to not count cells multiple times, some of the cells were removed. This was done in the following manner: if there was an area of overlap in multiple sections, only cells belonging to the sections with lower $x$ and $y$ positions were kept, for example in *Figure 3a*, any cells in the upper hypocotyl section that also belong spatially to the lower hypocotyl section, were removed from subsequent analysis. In the repeat WT experiment, the root tip section imaged encompasses a longer section of the root (*Figure 2—figure supplement 1c*). Hence, in order to make the analysis comparable to WT (*Figure 3*), we split the root tip section for further analysis. We considered the root tip cells of the repeat to be only those less than 0.18 mm from the actual tip, while the rest of them were classed as 'Root up from tip' (*Figure 2—figure supplement 1a,b*).

In the case of analysis at tissue level, where multiple sections had to be pooled for analysis (e.g. *Figure 1g,h* and *Figure 1—figure supplements 3*, *4* and *5*), since different sections were imaged at different times, before any further statistics were done, all the data were interpolated at the times where measurements across any section were made.

For analysis of amplitudes, peak and trough times for the individual cells (*Figure 1—figure supplement 6*; *Figure 2a,f,g*; *Figure 2—figure supplement 1a,e,f*; *Figure 2—figure supplement 2a,e, f*) were identified using the findpeaks function in MATLAB. This was done on linearly detrended data. In case of the WT data (*Figure 2*), since the data are sampled more frequently (every 1.1 vs. 3 hr in WT repeat and CCA1-long line), the data are noisier, hence a smoothing filter (robust local regression using weighted linear least squares and a Second degree polynomial model) was also

applied after linear detrending. Amplitudes of traces were calculated as a mean of all troughs to peak and peak to trough amplitudes. For amplitude calculations, only rhythmic cells that had at least three peaks detected during the movie were analysed. Also, if the number of peaks and troughs identified by the analysis differed by more than one the cell was discarded from the analysis.

## Luciferase space-time analysis

To facilitate analysis, individual seedlings were manually cropped into individual time stacks using ImageJ. From these image stacks a rectangular region of interest (ROI) containing the full length of the root and as much length of the hypocotyl as possible, whilst still excluding the cotyledons, was defined. Custom developed MATLAB scripts were used to extract luminescence data for each pixel in the ROI, giving time series for each pixel. After inspection of the images and the time series, some features were identified and the following measures applied to address them:

I.   Occasionally the cotyledon of the seedling or of a neighbouring seedling protrudes into the ROI. At this stage the ROI was checked for pixels of overlapping seedlings and these regions were manually removed from the affected frames.
II.  Inside of the ROI the hypocotyl and root are surrounded by peripheral background pixels. The root and hypocotyl were segmented from the background using the mean of the grey levels as the threshold. The algorithm was applied to each image in the stack individually.
III. Commonly supposed to be from solar cosmic rays, pixel spikes in intensity values occur sporadically in images. A 3-by-3 pixel median filter is applied to each image to remove these spikes.
IV.  The luminescence signal strength in a single seedling is weak and therefore the signal to noise ratio relatively low. A third order Butterworth filter was applied to pixel time series to remove high frequency noise. Time series were filtered using MATLAB's filtfilt.m function, which performs in the forward and reverse direction to avoid phase distortion. A cut off frequency of 15% of the Nyquist frequency was identified as a best fit to our data.
V.   In all experiments, we observed dampening of the signal over time. Time series were therefore amplitude de-trended to better visualise spatial patterns. Time series were de-trended using the algorithms developed for the mFourfit toolkit (*Edwards et al., 2010*).

To visualise spatial patterns across the length of the root, space-time plots of the root luminescence were created (*Figure 3i*, *Figure 3—figure supplement 1*). To do this, we take the maximum signal intensity across one pixel wide longitudinal sections of the root for each image and assign this value to position $m,n$ of the space time plot, where $m$ is the image number and $n$ the longitudinal section. The space-time plots presented include 10 pixels of the hypocotyl. The mean luminescence is normalised so that the peak expression of each longitudinal section ($n$) is 1.

## Model simulation

In *Figure 1a*, we simulate an existing deterministic model of the clock (*Pokhilko et al., 2012*). The model was run for 168 hr from introduction into constant light conditions and *LHY/CCA1* mRNA is reported (in the model *CCA1* and *LHY* are treated as a single component [*Pokhilko et al., 2012*]).

In *Figure 2*, we simulated a stochastic model of an existing circadian clock model (*Guerriero et al., 2012*; *Pokhilko et al., 2012*). In Guerriero et al, the model is scaled by the parameter Ω, so that a molecule count close to Ω is obtained. For detailed description of the scaling, the reader can refer to this paper. Comparison of the model simulated for different Ω values to the previously published data indicates that the model molecule count of a few hundred cells (i.e. Ω) is a good prediction of the actual molecule count (*Guerriero et al., 2012*). Here, we have taken the same circadian clock model and simulated it for various values of Ω. Model equations scaled for the Ω factor are given in (*Guerriero et al., 2012*). The model was simulated for 200 hr from introduction into constant light conditions and 100 simulation runs (proxy for 100 cells) were performed. The stochastic simulations were performed using the Gillespie algorithm (*Gillespie, 1977*). For each simulation, further analysis of amplitudes and period was done after the simulated data were interpolated at 2 hr intervals and then only for the simulated data from 28 hr to 168 hr in LL, in order to be closely comparable to the time interval of the original single cell data (*Figure 1d*). The Gillespie algorithm was written in MATLAB and the amplitudes and periods of the simulations were extracted using the MATLAB findpeaks function. Periods were calculated as a mean difference of peak-to-peak intervals. Amplitudes were calculated as a mean of all trough to peak and peak to trough amplitudes.

## Synchronisation analysis

For a set of individual cells, the inter-cellular synchrony was analysed. First, one cell was selected as a centroid of the synchronisation analysis. Then, its neighbouring cells, defined as those located within its sphere (radius $r$), were extracted. From CCA1-YFP expression signal, phase of the $j$-th neighbouring cell ($j$ = 1,2..,N) was computed as (*Pikovsky et al., 2001*)

$$\theta_j(t) = 2\pi\, k + \frac{t - t_k}{t_{k+1} - t_k} \times 2\pi.$$

Here, the $k$-th peak time $t_k$ of the bioluminescence signal was detected by a cosine fitting method (coefficient of determination larger than 0.7) using the estimated period $\tau_i$. Then for each time point, the order parameter $R(t)$ (*Kuramoto, 1984*) was obtained as

$$R(t)\, e^{\Theta} = \frac{1}{N} \sum_{j=1}^{N} e^{-i\theta_j(t)}$$

The order parameter ($0 < R < 1$) becomes unity for completely synchronised cells ($\theta_1 = \theta_2 = .. = \theta_N$), whereas it becomes zero for non-synchronised cells. *Figure 3—figure supplement 2* shows the results of synchronisation analysis for root tip (a), lower root (b), upper root (c), lower hypocotyl (d), upper hypocotyl (e), and cotyledon (f). For each section, a total of $n$ curves were drawn by selecting individual cells as the centroids (root tip, $n$ = 242; lower root, $n$ = 84; upper root, $n$ = 46; lower hypocotyl, $n$ = 114; upper hypocotyl, $n$ = 53; cotyledon, $n$ = 103). By linear regression analysis of each curve, the slope of the order parameter against time was computed, where a positive slope implies that the level of synchrony increases in time due to cell-to-cell interactions. *Figure 3—figure supplement 2h* shows the dependence of the slope value on cell density ($\rho = N/(4/3)\pi r^3$). Positive slopes are mostly found in the root tip (*Figure 3—figure supplement 2g*), with a high correlation to the cell density.

Next, the coupling strength was estimated for each synchronisation curve $\{R(t)\}$. Our approach is based upon a simplified version of the technique developed for weakly interacting mammalian circadian cells (*Rougemont and Naef, 2007*). As a model for the neighbouring cells, we consider a set of coupled phase oscillators

$$\frac{d\theta_j}{dt} = \omega_j + \frac{K}{N} \sum_{k=1}^{N} \sin\left(\theta_k - \theta_j\right)$$

Assuming that the period $\tau_j$ estimated from the $j$-th cellular trace is not strongly affected by the other cells (*Rougemont and Naef, 2007*), the natural angular frequency was set as $\omega_j = 2\pi/\tau_j$ for each oscillator. Given an initial condition $\theta_j(0)$ extracted from the cellular traces, the phase oscillator model was simulated (Euler method with time step 0.1 hr). Accordingly, the time evolution of the order parameter $R(t)$ could be obtained. The coupling strength, which was initially set as $K$ = 0.002, is constant for each simulation. Staring from the minimum level of coupling, the coupling strength was slowly increased so that the phase oscillators are eventually mutually synchronised and the corresponding slope value increases monotonously. At the point when the slope value exceeds the one obtained from the experiment, the corresponding value of $K$ provides the coupling estimate for the experimental data. *Figure 3—figure supplement 2i* shows the results. Stronger coupling was estimated for densely populated areas, implying that the cell-to-cell interactions are strengthened when cells are closely located to each other.

To examine the dependence of the present analysis on the synchrony measure used, the synchronisation index (Garcia-Ojalvo, Elowitz, Strogatz, 2004) was utilized in place of the order parameter. The synchronisation index has the advantage that the noise-sensitive procedure of phase extraction from the cellular traces is not required, since it can be computed directly from the measured signals. For $N$ cellular traces $\{x_j(t): j = 1,2,..,N\}$, the averaged signal $M(t) = (1/N) \Sigma_j x_j(t)$ is computed. Then the synchronisation index is given by

$$R = \frac{\langle M^2 \rangle - \langle M \rangle^2}{(1/N) \sum_j^N \left\{ \langle x_j^2 \rangle - \langle x_j \rangle^2 \right\}},$$

where <> denotes time average. In a synchronised cellular state, the averaged signal gives rise to a pronounced amplitude, resulting in $R$=1. The fully desynchronised cellular state, on the other hand, results in $R$=0. To see the time evolution of the level of synchrony, the synchronisation index $R$ ($t$) at time $t$ was computed for windowed time traces of $\{x_j(s): t-12<s<t+12\}$ (window: 24 h). *Figure 3—figure supplement 3* shows the analysis results based on the synchronisation index. The panels are ordered in correspondence with those of *Figure 3—figure supplement 2*. Positive slopes are again found in the root tip, implying that the level of synchrony increases in time due to cell-cell interactions. The results are therefore consistent with the ones obtained by the order parameter.

To examine the dependence of the synchronisation analysis on the particular experimental data set used, the order parameters were computed for the CCA1-long line. As shown in *Figure 3—figure supplement 4*, the slope of the order parameter for the CCA1-long line experiment is again well correlated with the cell density, where highly dense cells are located in the root tip. Although the time resolution was three times lower in this experiment, the same tendency was observed. The WT repeat experiment (*Figure 2—figure supplement 1*) was not analysed, as the timeseries was too short and time resolution was too low to enable accurate synchronisation analysis.

## Phase oscillator model

We constructed a model where we describe the dynamics of the CCA1::CCA1-YFP in each cell by a simple Kuramoto phase oscillator. For every cell at a position ($m$,$n$) in the plant (when viewed in 2D with $m$ denoting position in the horizontal direction and $n$ denoting position in the vertical direction) the phase of the oscillator $\theta^{(m,n)}$ changes in time ($t$) so that

$$\frac{d\theta^{(m,n)}}{dt} = \omega^{(m,n)} + K \sum_{\langle p.q \rangle} \sin\left(\theta^{(p,q)} - \theta^{(m,n)}\right).$$

Here $\omega^{(m,n)}$ describes the intrinsic frequency of the oscillator and the second term describes the oscillator's dependence on the coupling to the nearest neighbours (i.e. cells in positions ($p$,$q$) where $p$=$m$-1,..,$m$+1 and $q$=$n$-1,. .,$n$+1) with $K$ as the coupling constant. The bioluminescence of each cell is then taken to be $B^{(m,n)}(t) = \cos\left(\theta^{(m,n)}(t)\right) + 1$. Cells in the plant conform to a template that is taken to be a symmetric shape, and resembles the shape of a seedling. The total bioluminescence across each vertical section ($n$) of the plant is taken to be the sum of the bioluminescence of all cells along that section that is, $B_{tot}^n(t) = \sum_{m=1}^{Nw} B^{(m,n)}(t)$ where the width of the plant counts $Nw$ number of cells. The space-time plot shown in *Figure 3k* shows the total luminescence normalised so that the peak expression of each vertical section is 1.

In the model, we assume that the cells in the three sections (the cotyledon/hypocotyl, the root and the root tip) have different intrinsic periods, with the cells in the cotyledon and hypocotyl having period of 24 hr, those in the root having the period of around 25.55 hr and the ones in the root tip a period of 22.67 hr. These overall match the qualitative period differences seen across the different plant sections. In all simulations, the coupling constant $K$ is arbitrarily set to 1. The ODEs are solved using the Euler method.

## Data availability

Single cell datasets generated in this study are available at https://gitlab.com/slcu/teamJL/Gould_etal_2018 (*Gould et al., 2018*); copy archived at https://github.com/elifesciences-publications/teamJL/Gould_etal_2018). Single cell datasets are also available at the University of Cambridge Data repository at https://doi.org/10.17863/CAM.23042. MATLAB code for the phase oscillator model and key data analysis code are also available on the GitLab page. Some of the figures were produced using packages 'Controllable Tight Subplot' and 'Red Blue Colormap' available from Matlab File Exchange.

## Acknowledgements

We would like to acknowledge the Liverpool Centre for Cell Imaging for assistance and maintenance of the confocal microscope, specifically Marco Marcello facility manager and David Mason image analysis support. AH and PDG were funded by BBSRC grant BB/K018078/1, JCWL and MD by BBSRC grant BB/K017152/1 and MG by the Liverpool/Durham/Newcastle BBSRC DTP.

# Additional information

## Funding

| Funder | Grant reference number | Author |
|---|---|---|
| Biotechnology and Biological Sciences Research Council | BB/K018078/1 | Peter D Gould Anthony JW Hall |
| Biotechnology and Biological Sciences Research Council | BB/K017152/1 | James CW Locke Mirela Domijan |
| Biotechnology and Biological Sciences Research Council | Liverpool/Durham/ Newcastle BBSRC DTP | Mark Greenwood |
| Gatsby Charitable Foundation | | James CW Locke |
| H2020 European Research Council | | James CW Locke |

The funders had no role in study design, data collection and interpretation, or the decision to submit the work for publication.

## Author contributions

Peter D Gould, Conceptualization, Data curation, Formal analysis, Validation, Investigation, Methodology, Writing—original draft, Writing—review and editing; Mirela Domijan, Conceptualization, Data curation, Software, Validation, Investigation, Methodology, Writing—original draft, Writing—review and editing; Mark Greenwood, Data curation, Software, Validation, Investigation, Methodology, Writing—original draft, Writing—review and editing; Isao T Tokuda, Software, Investigation; Hannah Rees, Investigation; Laszlo Kozma-Bognar, Resources, Investigation; Anthony JW Hall, Conceptualization, Resources, Supervision, Funding acquisition, Validation, Methodology, Writing—original draft, Writing—review and editing; James CW Locke, Conceptualization, Supervision, Funding acquisition, Writing—original draft, Project administration, Writing—review and editing

## Author ORCIDs

Mirela Domijan (iD) http://orcid.org/0000-0003-3853-9119
Isao T Tokuda (iD) http://orcid.org/0000-0001-6212-0022
Laszlo Kozma-Bognar (iD) http://orcid.org/0000-0002-8289-193X
James CW Locke (iD) http://orcid.org/0000-0003-0670-1943

## Decision letter and Author response

Decision letter https://doi.org/10.7554/eLife.31700.033
Author response https://doi.org/10.7554/eLife.31700.034

# Additional files

## Supplementary files

• Transparent reporting form
DOI: https://doi.org/10.7554/eLife.31700.025

## Data availability

Single cell data is available from https://gitlab.com/slcu/teamJL/Gould_etal_2018

The following datasets were generated:

| Author(s) | Year | Dataset title | Dataset URL | Database, license, and accessibility information |
|---|---|---|---|---|
| Gould PD, Domijan M, Greenwood M, Tokuda IT, Rees H, Kozma-Bognar L, Hall AJW, Locke | 2018 | WT | https://gitlab.com/slcu/teamJL/Gould_etal_2018/tree/master/Single-CellFiles/Data_singlecell/WT_final_coordinates | Publicly available at GitHub (repository https://gitlab.com/slcu/teamJL/Gould_etal_2018) |

JCW

| Gould PD, Domijan M, Greenwood M, Tokuda IT, Rees H, Kozma-Bognar L, Hall AJW, Locke JCW | 2018 | WT repeat | https://gitlab.com/slcu/teamJL/Gould_etal_2018/tree/master/SingleCellFiles/Data_singlecell/WTrepeat_final_coordinates | Publicly available at GitHub (repository https://gitlab.com/slcu/teamJL/Gould_etal_2018) |
|---|---|---|---|---|
| Gould PD, Domijan M, Greenwood M, Tokuda IT, Rees H, Kozma-Bognar L, Hall AJW, Locke JCW | 2018 | CCA1-Long | https://gitlab.com/slcu/teamJL/Gould_etal_2018/tree/master/SingleCellFiles/Data_singlecell/CCA1-long_final_coordinates | Publicly available at GitHub (repository https://gitlab.com/slcu/teamJL/Gould_etal_2018) |

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
