## [Decision Letter]

Thank you for submitting your article "Coordination of robust single cell rhythms in the Arabidopsis circadian clock via spatial waves of gene expression" for consideration by *eLife*. Your article has been reviewed by two peer reviewers, and the evaluation has been overseen by Richard Amasino as the Reviewing Editor and Christian Hardtke as the Senior Editor. The following individual involved in review of your submission has agreed to reveal his identity: Andrew Millar.

The Reviewing Editor has drafted the comments below, which are based on the reviews, to help you prepare a revised submission.

Two papers from Tokitaka Oyama (Okada et al. 2017 and Muranaka and Oyama, 2016) have done fairly significant scale of single-cell imaging, both in terms of number of cells and of time series duration. Although these were using Lemna fronds, and this Arabidopsis work addresses considerably greater organ complexity, these two papers ought to be cited.

The authors argue that damping in individual cells is small relative to cell desynchrony, largely it seems by inspection of Figure 1—figure supplement 5. That argument is reasonable – the conclusions of the present paper depend more on the clear differences among tissues than the more subtle differences among cells within a tissue. Future work will require a more objective method to score damping in individual traces, which should be possible from individual peak/trough data.

The model result in Figure 3K achieves a qualitative match to the spatial pattern of phases along the root, driven simply by period differences of 1.5 to 3h among root sections. One difficulty inherent in the modelling is that these period differences are estimated from intact plants, so the measured periods include effects of coupling. This is unavoidable, as there is currently no reliable means to measure the rhythmic properties of physiologically-representative plant cells in physical isolation, and this issue needs to be clearly noted in the paper.

"waves cause the most desynchronisation." Averaging over the waves of mean phase along the y axis is one factor that contributes to damped rhythms in the root as a whole. The other factor is the more variable period among cells at each y position in the root (e.g. Figure 3H), compared to the hypocotyl or cotyledon, which the authors attribute to weaker coupling. These two sources of desynchrony were not quantitatively compared, and preferably they should be. Alternatively, the authors could qualify the claim that averaging of the mean phase along the root has a greater effect than desynchrony at a single location.

To convey the scale of the rhythmic patterns and the difference among locations, from the data analysis here, can the authors estimate a characteristic separation distance where clocks have similar phases, compared to more distant clocks?

Absence of radial (z-axis) patterning in the root (Results, fifth paragraph) is a new and interesting result, suggesting that rhythmic properties are not cell-type specific in the bulk of the root. This is the first dataset with both the spatial resolution and temporal duration to address this issue for any plant organ. However, only one dataset is shown with z-axis resolution. Does the same result hold in the repeat seedlings?

Data and code availability:

It is good to release the numerical data on BioDare, but BioDare hosts many public datasets. The authors should give specific identifiers for each experiment, data set or script in the paper, perhaps as a table, to relate specific figures to the relevant, underlying data. BioDare Experiment IDs are unique, stable identifiers for this specific purpose. Will the fluorescence image stacks also be available? Preferably, however, the data ought to be deposited at Dryad (http://datadryad.org/) because this site is likely to be more stable long-term.

The periods of the clock in the model were faster in the shoot and root tip thanthe rest of the root. Might this be related to patterns of somite rhythms (Benfey lab papers) that are associated with lateral root emergence. If the plant clock drives auxin rhythms, are these causal to developmental outcomes at the single-cell level? Although outside the scope of this work, maybe a sentence of speculation in the Discussion might be helpful to a general audience?

[Editors' note: further revisions were requested prior to acceptance, as described below.]

Thank you for resubmitting your work entitled "Coordination of robust single cell rhythms in the Arabidopsis circadian clock via spatial waves of gene expression" for further consideration at *eLife*. Your revised article has been favorably evaluated by Christian Hardtke as the Senior Editor, Richard Amasino as the Reviewing Editor, and one reviewer.

It has long been observed that plant rhythms damp over time. This has been attributed to desynchronizing of oscillators in individual cells, but there has been little experimental support for this hypothesis. Your work provides such experimental evidence. It also describes waves of clock gene expression running from shoot to root and from the root tip up the root which strongly supports coupling among clocks in different cells. This work is likely to stimulate and inform the next rounds of experiments and models. The manuscript has been improved but there are some remaining major points that must be addressed:

Introduction, third paragraph. Figure 1C is described as showing individual cells losing rhythmicity, but doesn't it actually show all individual cells losing amplitude, which is different. If individual cells lose rhythmicity would you not have some cells remaining rhythmic (with or without a loss in amplitude) while other cells cease cycling and instead display either constant expression or randomly varying expression levels?

Results, third paragraph. The authors state that "most" cells display circadian rhythms. Can this be quantified? Is most 51% or 90%? Although this would be somewhat arbitrary, based on the cutoffs chosen, it is important to provide the reader with an estimate in the text.

Results, fourth paragraph. How was it determined that the single cell rhythms in the hypocotyl and cotyledons are more robust than a system size of 100 and are closer to a system size of 1000? Was this by visual inspection? Can this be quantified? Perhaps through a comparison of the variances of estimates of period and phase? Similarly, is the greater variability of root phases than phases in the rest of the plant (Results, fifth paragraph) by inspection? Could this be quantified?

Results, last paragraph. Cell density is greater in the root tip than elsewhere, which may contribute to increased coupling. Another tissue with increased cell density is the Shoot Apical Meristem. Given the suggestion of Takahashi et al. that the SAM is important for the maintenance of root rhythms, it would be interesting to look at rhythmicity in the SAM. Although an examination of the SAM is not necessary for this manuscript, the authors may want to consider this and mention it in the paper.

---

## [Author Response]

Two papers from Tokitaka Oyama (Okada et al. 2017 and Muranaka and Oyama, 2016) have done fairly significant scale of single-cell imaging, both in terms of number of cells and of time series duration. Although these were using Lemna fronds, and this Arabidopsis work addresses considerably greater organ complexity, these two papers ought to be cited.

We agree that these important papers should be cited. We have added these references to the Discussion and describe their results (second paragraph).

The authors argue that damping in individual cells is small relative to cell desynchrony, largely it seems by inspection of Figure 1—figure supplement 5. That argument is reasonable – the conclusions of the present paper depend more on the clear differences among tissues than the more subtle differences among cells within a tissue. Future work will require a more objective method to score damping in individual traces, which should be possible from individual peak/trough data.

We agree that analysis of peak/trough data would be informative. To address this, we have generated a new supplemental figure (Figure 1—figure supplement 6) examining peak/trough amplitudes to address this point from our data.

The model result in Figure 3K achieves a qualitative match to the spatial pattern of phases along the root, driven simply by period differences of 1.5 to 3h among root sections. One difficulty inherent in the modelling is that these period differences are estimated from intact plants, so the measured periods include effects of coupling. This is unavoidable, as there is currently no reliable means to measure the rhythmic properties of physiologically-representative plant cells in physical isolation, and this issue needs to be clearly noted in the paper.

We agree that this is an important, although unavoidable, caveat to the modelling. We now clearly note this in the second paragraph of the Discussion. However, we note that the modelling sets out to show that (i) intrinsic period differences (fast-slow-fast) and (ii) weak coupling are sufficient ingredients for creating a wave. At this stage, the actual period values we subscribe are not critical for this qualitative behaviour.

"waves cause the most desynchronisation." Averaging over the waves of mean phase along the y axis is one factor that contributes to damped rhythms in the root as a whole. The other factor is the more variable period among cells at each y position in the root (e.g. Figure 3H), compared to the hypocotyl or cotyledon, which the authors attribute to weaker coupling. These two sources of desynchrony were not quantitatively compared, and preferably they should be. Alternatively, the authors could qualify the claim that averaging of the mean phase along the root has a greater effect than desynchrony at a single location.

We agree with the reviewer that the desynchronisation caused by the more variable periods observed in the root is a significant cause of the damping observed at the tissue level and should be discussed. We have now qualified our claims in the last paragraph of the Introduction and in the sixth paragraph of the Results, as well as characterised and discussed the desynchronisation that we observe.

To convey the scale of the rhythmic patterns and the difference among locations, from the data analysis here, can the authors estimate a characteristic separation distance where clocks have similar phases, compared to more distant clocks?

We agree with the reviewer that a characteristic separation distance would be an interesting metric to examine the scale of the rhythmic patterns. Unfortunately, due to the differences in synchronicity and wave speed in the different tissues a single separation distance is not possible. We have however added, in response to another comment (see below), the average phase and the distribution of phases for separate sections of the plant. This allows the reader to see the difference between locations of the clock rhythm.

Absence of radial (z-axis) patterning in the root (Results, fifth paragraph) is a new and interesting result, suggesting that rhythmic properties are not cell-type specific in the bulk of the root. This is the first dataset with both the spatial resolution and temporal duration to address this issue for any plant organ. However, only one dataset is shown with z-axis resolution. Does the same result hold in the repeat seedlings?

Thank you for this excellent suggestion. The result does hold for the repeat seedlings. We have added a new supplemental figure (Figure 2—figure supplement 3) to show this, and have added references to it in the text.

Data and code availability:It is good to release the numerical data on BioDare, but BioDare hosts many public datasets. The authors should give specific identifiers for each experiment, data set or script in the paper, perhaps as a table, to relate specific figures to the relevant, underlying data. BioDare Experiment IDs are unique, stable identifiers for this specific purpose. Will the fluorescence image stacks also be available? Preferably, however, the data ought to be deposited at Dryad (http://datadryad.org/) because this site is likely to be more stable long-term.

We have had recent difficulty accessing Biodare, so to make the data easily available we have added the single cell data sets on the Sainsbury Laboratory’s publically available GitLab page (https://gitlab.com/slcu/teamJL/Gould_etal_2018), with clear references to the data. The benefit of placing the data on GitLab is that we have also added many of the analysis files, allowing users to recreate multiple figures from the paper. On acceptance, we will also add the single cell data sets to the Cambridge Data Repository

(https://www.repository.cam.ac.uk/). We hope that the combination of the two sources should maximise the ease for researchers to examine our results and access our data.

The periods of the clock in the model were faster in the shoot and root tip thanthe rest of the root. Might this be related to patterns of somite rhythms (Benfey lab papers) that are associated with lateral root emergence. If the plant clock drives auxin rhythms, are these causal to developmental outcomes at the single-cell level? Although outside the scope of this work, maybe a sentence of speculation in the Discussion might be helpful to a general audience?

We have added a sentence on this possibility to the Discussion, as suggested.

[Editors' note: further revisions were requested prior to acceptance, as described below.]

[…] The manuscript has been improved but there are some remaining major points that must be addressed:Introduction, third paragraph. Figure 1C is described as showing individual cells losing rhythmicity, but doesn't it actually show all individual cells losing amplitude, which is different. If individual cells lose rhythmicity would you not have some cells remaining rhythmic (with or without a loss in amplitude) while other cells cease cycling and instead display either constant expression or randomly varying expression levels?

We agree that the phrase ‘losing rhythmicity’ does not precisely describe the traces drawn in Figure 1C. We have rewritten the text to be:

‘This damping at the whole plant level could be due to the clock circuit in all individual cells damping (top, Figure 1C), or to cells desynchronising due to different intrinsic periods or phases’

Results, third paragraph. The authors state that "most" cells display circadian rhythms. Can this be quantified? Is most 51% or 90%? Although this would be somewhat arbitrary, based on the cutoffs chosen, it is important to provide the reader with an estimate in the text.

The number of cells that displayed circadian rhythms in each section of the plant is quantified in the Figure 1—source data 1 and 2, which were unfortunately omitted from the resubmission. I am sorry for the confusion caused by this mistake. These are now included, and we have simplified the tables to make them easier to understand. We also now describe the tables in the main text (Results, third paragraph).

Results, fourth paragraph. How was it determined that the single cell rhythms in the hypocotyl and cotyledons are more robust than a system size of 100 and are closer to a system size of 1000? Was this by visual inspection? Can this be quantified? Perhaps through a comparison of the variances of estimates of period and phase?

The statement that the hypocotyl and cotyledon are more robust than a system size of 100 was based on the comparison of the mean trace amplitude versus the median amplitudes of the individual traces for experiment (Figure 2A) and model at different system sizes (Figure 2D) and the comparison of the period variability for the experiment (Figure 2B) and the model (Figure 2E). We agree that this was not clear from the way the text was written. We have now clarified this in the main text (Results, fourth paragraph).

Similarly, is the greater variability of root phases than phases in the rest of the plant (Results, fifth paragraph) by inspection? Could this be quantified?

We described in the text the standard deviations of the phase distributions of the different tissues. These get larger in the root and root tip.

‘Examination of the clock phase between 48h to 72h reveals that the clock peaks earlier in the hypocotyl (mean phase in upper hypocotyl 51.2h ± 1.1h, mean phase in lower hypocotyl 52.1h ± 1.5h) than in the cotyledon (mean phase 55.6h ± 2.3h) (Figure 3F-H). […] However, from the root tip (mean phase 56.2h ± 5.0h) the phase of the clock is shifted to later in the day as you go up (lower root phase 58.0h ± 4.4h).’

To make the comparison of the variability in the root to the rest of the plant clearer, we have modified the text to state:

‘At each y position in the root the phases are also more variable than in the rest of the plant, with a standard deviation of phases of ± 5.0h in the root tip compared to ± 1.1h in the hypocotyl for phases occurring between 48h and 72h (Figure 3H).’

Results, last paragraph. Cell density is greater in the root tip than elsewhere, which may contribute to increased coupling. Another tissue with increased cell density is the Shoot Apical Meristem. Given the suggestion of Takahashi et al. that the SAM is important for the maintenance of root rhythms, it would be interesting to look at rhythmicity in the SAM. Although an examination of the SAM is not necessary for this manuscript, the authors may want to consider this and mention it in the paper.

We agree that this is an interesting point for future work and have added a sentence on this in the text:

‘In the future, it will be interesting to examine whether the shoot apical meristem, which has a high cell density like the root tip, has a similarly high coupling strength given its role in driving rhythms in the shoot (Takahashi et al., 2015).’